# An Ounce of Prevention, a Pound of Complications: A Case of Statin-Induced Necrotizing Myopathy in a Frail Elderly Patient

**DOI:** 10.3390/geriatrics7020033

**Published:** 2022-03-18

**Authors:** Oleg Stens, Bradley Neutel, Elizabeth L. Goodman

**Affiliations:** 1Department of Medicine, University of California, San Diego, CA 92161, USA; ostens@health.ucsd.edu; 2Department of Medicine, Harbor-UCLA Medical Center, Torrance, CA 90502, USA; bneutel2@dhs.lacounty.gov

**Keywords:** statins, primary prevention, stain-induced necrotizing myopathy, anti-HMG co-reductase myopathy

## Abstract

The use of statins for primary prevention in older adults remains controversial. In this manuscript, we present a case of an 81-year-old woman with a history of HTN, HLD, Alzheimer’s dementia and osteoporosis, who presented to a geriatrics clinic with profound muscle weakness accompanied by new functional deficits in the setting of taking double her prescribed dose of atorvastatin. She was admitted to the hospital where she was found to have rhabdomyolysis. Muscle biopsy and serologic work up revealed anti-HMG statin co-reductase myopathy as the cause of her symptoms. The patient was treated with steroids IVIG and immunomodulators with marked improvement in her weakness; however, her course was complicated by delirium and multiple falls, resulting in several fragility fractures. This case highlights the need to conduct a risk–benefit analysis prior to initiating new therapies in patients with limited life expectancy, including the consideration of the potential for medication errors.

## 1. Introduction

The pleotropic effects on statins on lipids and atherosclerotic plaque stabilization have proven to be a cornerstone in the prevention of cardiovascular events and death. It is well-established that the use of statins in patients 75 or younger with elevated risk of atherosclerotic cardiovascular disease (ASCVD) reduces the relative risk of major cardiovascular events by 20–30% [1]. However, a paucity of data exists regarding primary prevention in patients older than 75. Although the United States Preventative Task Force (USPTF) and several other consensus guidelines have concluded there is insufficient evidence to either recommend or discourage the use of statin in adults greater than 75 for primary prevention [2], the rates of statin use for primary prevention in adults older 79 have increased three-fold during the first decade of this century [3]. This case demonstrates an unusual side effect in a frail elderly patient initiated on a statin for primary prevention.

## 2. Case Report

An 81-year-old woman, with a history of hypertension, dyslipidemia, T12 compression fracture and Alzheimer’s dementia, presented to established care at a geriatric clinic with a chief complaint of two months of gradual progressive symmetric proximal muscle weakness. At baseline, the patient was functionally dependent only on her instrumental activities of daily living. However, her severe weakness resulted in new functional deficits including difficulties in standing, walking and transferring. On medication reconciliation, the patient was noted to be taking atorvastatin 40 mg from two different bottles, doubling her daily dose. Atorvastatin had been started for primary prevention at the age of 79 for an atherosclerotic cardiovascular disease (ASCVD) risk score of >7%. Additional medications at the time of her clinic visit included amlodipine 5 mg daily, donepezil 5 mg nightly, memantine 10 mg twice a day, pantoprazole 40 mg a day and a calcium/Vitamin D supplement. Her exam was significant for 3/5 strength in hip flexors, resulting in the inability to stand unassisted and an inability to abduct her arms past 90 degrees. She was started on prednisone 15 mg orally a day for a presumed diagnosis of polymyalgia rheumatica, and her statin was discontinued. Two weeks later, she was seen in follow-up in the geriatric clinic and found to have profound weakness in the upper and lower extremities as well as new dysphagia and mild dysarthria. She was admitted to the hospital where her workup was notable for mildly elevated troponin-I, elevated AST and ALT and a creatine kinase (CK) of 7630 U/L (normal 38–234 U/L). A urine dipstick showed 3 + blood with only one RBC/high powered field on microscopy. Creatinine was within normal limits. Electromyography (EMG) showed a proximal greater than distal irritable myopathy. A muscle biopsy of the left quadriceps showed segmental necrosis of skeletal muscle fascicles with perivascular infiltration by T-cells, B-cells and plasma cells, and lipid droplet accumulation was consistent with an immune-mediated necrotizing myopathy causing rhabdomyolysis (Figure 1, Figure 2, Figure 3 and Figure 4). Anti-HMG CoA reductase antibodies were positive at >200 units (normal <20 units), whereas ANA, anti-Jo, MI-2, SRP, RNP, Smith, Scl-70 and SS-A/SS-B antibodies all were negative.

The patient was given intravenous hydration for her rhabdomyolysis. She was started on IVIG infusion for 5 days and then initiated on a prednisone taper. The patient’s speech and swallowing improved and she was able to tolerate thin liquids with no aspiration. Her CK levels decreased five-fold over the course of two weeks, her elevated liver enzymes resolved and her troponin decreased (Table 1). She was discharged to a skilled nursing facility with rehabilitation with outpatient IVIG infusions, mycophenolate mofetil and a methylprednisolone taper.

The patient was ultimately readmitted twice in the following year, both times for ground level falls, which resulted in bilateral intertrochanteric fractures treated with intermedullary nailing and a left proximal phalangeal fracture treated non-operatively. As of her last follow-up, the patient remains home-dwelling with an in-home caregiver. Her proximal muscle strength is preserved, and she ambulates with a four-wheel walker. Her dementia has progressed to Fast Stage 6d. Her caregiver reported that she continues to dance at church.

At her five month follow-up visit, the patient was found to have near-normal strength and was participating in a dancing program at her adult day care. Her IVIG infusions were terminated early due an association with delirium along with the rapid improvement in her strength. Her steroids tapered, and mycophenolate was continued for a total of 10 months of treatment.

## 3. Discussion

The differential for the progressive weakness of proximal muscles in an elderly patient includes both PMR and myopathies. Myopathies can usually be readily differentiated from PMR in that CK is generally elevated. Patients with PMR should not have objective weakness; rather, they typically present with pain limiting their movements [4,5]. In this case, the acute onset and rapid decline of proximal muscle strength in the setting of a markedly elevated CK was most consistent with myopathy. Ultimately, the final diagnosis of statin-induced immune-mediated necrotizing myopathy was cinched by the markedly elevated anti-HMG CoA reductase antibody in combination with characteristic findings on muscle biopsy. Anti-HMG CoA reductase antibodies can be present in the absence of a statin exposure; however, most cases are associated with statin use, particularly those observed in older patients. While the risk of myotoxic side effects of statins have typically been associated with higher potency statins, a dose–response relationship with respect to statin-induced immune-medicated necrotizing myopathy has not been well established [6].

Data regarding the utilization of statin for primary prevention of ASCVD in older patients remain mixed. A large meta-analysis found an overall survival benefit from statin use in patients older than 75 [7]. In addition, a retrospective cohort study conducted by the US Veterans Health Administration (VHA) found that the initiation of statins for primary prevention in patients age 75 years or older was associated with a statistically significant reduction in the risk of all-cause and cardiovascular mortality [8].

On the other hand, a large retrospective trial of patients older than 74 without diabetes found no cardiovascular or all-cause mortality benefits for primary prevention with a statin [9], while a post hoc analysis of the ALLHAT-LLT trial, suggested a nonsignificant increase in mortality in older patients treated with pravastatin when compared to usual care [10]. Many of these trials had significant limitations and the hope is that the ongoing Statins in Reducing Events in the Elderly (STAREE) trial, a randomized controlled trial exploring the benefits of statins in patient’s over the age of 70, will provide a more definite understanding of the risks and benefits of statin use in this age group [11].

In the absence of high-quality data, it is not surprising that guidelines on statin use for primary prevention over the age of 75 are very heterogeneous. The United States Preventative Task Force (USPTF) 2016 guidelines and the American College of Cardiology/American Heart Association (ACC/AHA) 2019 conclude that there is insufficient evidence to support the initiation of a statin for primary prevention in adults older than 75 [12,13,14]. ACC/AHA states that providers may consider the initiation of a moderate-intensity statin over the age of 75l; however, they recommend stopping when the patient experiences decline in physical or cognitive function or if the patient has a coronary calcium score of zero [13]. The Canadian Cardiovascular Society (CCS) 2016 guidelines note that the indication for primary prevention with statins is poorly defined over the age of 75. Similarly, the European Society of Cardiology/European Atherosclerosis Society 2016 guidelines note that their risk scoring system is not intended for adults over 65 but recommend the consideration of statin if other risk factors such as smoking, diabetes, hyperlipidemia or hypertension are present.

On the other hand, the US Department of Veteran’s Affairs and Department of Defense 2014 guidelines and the National Institutes of Care and Health Excellence (NICE) 2014 guidelines both recommend initiation in patients age 75–84 depending on the risk level [12]. Reasons why statin initiation in older age may be less beneficial are complex. Not only do many older adults have life expectancies that are too short to experience the long-term benefits of statin use, they are also more likely to experience statin side effects. One study found that statin users over the age of 65 have over four-fold the risk of hospitalization for rhabdomyolysis compared to younger statin users [15]. Older patients are often on multiple medications, increasing the chances of drug–drug interactions and side effects, while patients with cognitive disorders are at higher risk of medication errors.

Given the current state of evidence for statin use for primary prevention in the elderly as well as the increased risk, it is our recommendation that overall life expectancy and the goals of care drive this decision-making process. This is consistent with the ABIM Choosing Wisely initiative, which recommends against routinely prescribing lipid-lowering medications in patients with limited life expectancy. In this case, the patient’s Alzheimer’s, dementia and other co-morbidities limited her life expectancy and increased the risk of a medication error. The weakness from myositis along with her cognitive deficits increased her risk of falls, while her pre-existing severe osteoporosis and the steroids needed to treat her condition left her even more vulnerable to fragility fractures. It is interesting to consider whether this cascade of events would have occurred had the patient never been initiated on a statin.

## Figures and Tables

**Figure 1 geriatrics-07-00033-f001:**
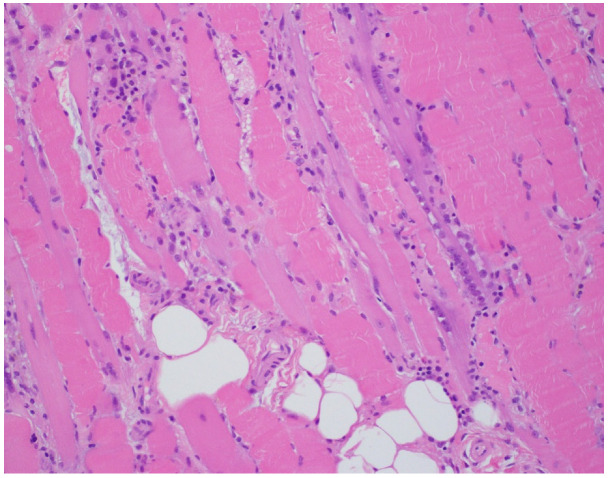
Hematoxylin and eosin stain: segmental necrosis and inflammatory infiltrate without vessel wall destruction.

**Figure 2 geriatrics-07-00033-f002:**
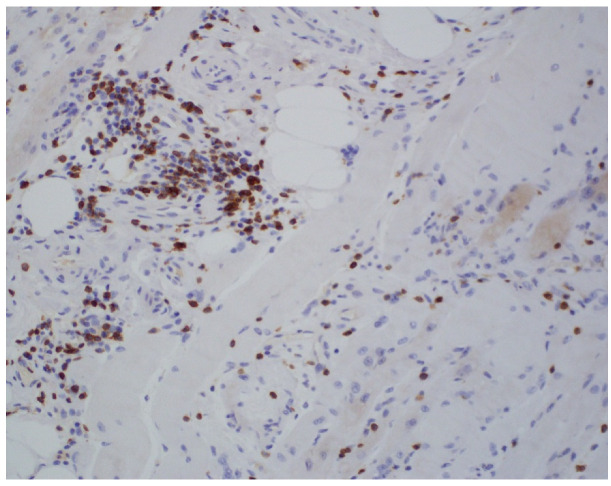
Immunohistochemistry (CD 68-macrophage) foci of infiltrate.

**Figure 3 geriatrics-07-00033-f003:**
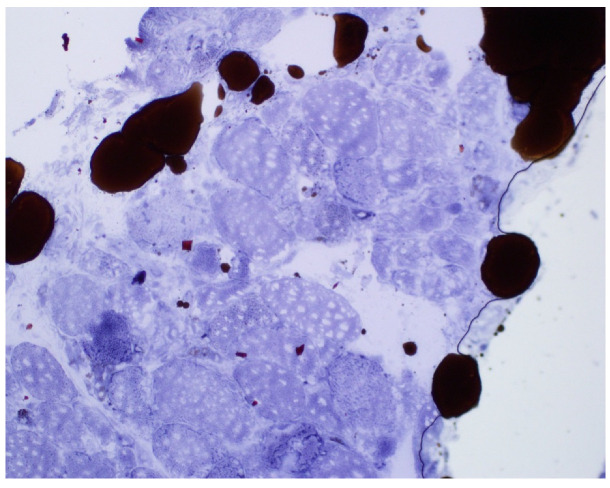
Sudan black stain: lipid droplets in scattered and intact myofiber (low power).

**Figure 4 geriatrics-07-00033-f004:**
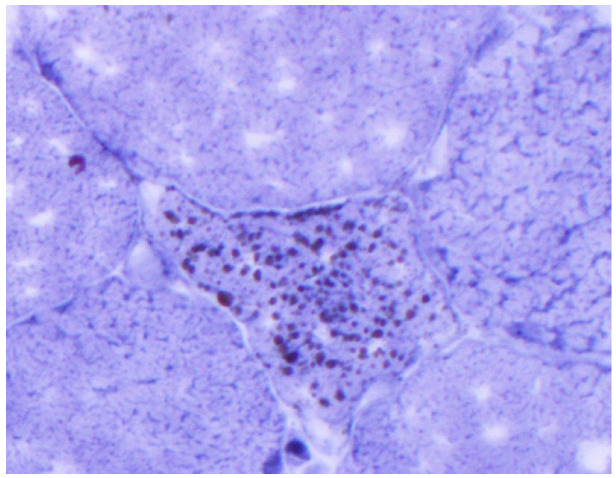
Sudan black stain: lipid droplets in scattered and intact myofiber (high power).

**Table 1 geriatrics-07-00033-t001:** Laboratory data.

Lab	Admission	Discharge	2 Months Post Discharge	8 Months Post Discharge
CK (U/L)	7628	1394	252	63
AST (U/L)	253	125	48	14
ALT (U/L)	319	162	59	8

## Data Availability

Not applicable.

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
