# Peer review of "An Ounce of Prevention, a Pound of Complications: A Case of Statin-Induced Necrotizing Myopathy in a Frail Elderly Patient"

_geriatrics, 2022, doi:10.3390/geriatrics7020033_

Round 1
Reviewer 1 Report
This case report emphasizes high value care in older adults.
Be sure to reference the ABIM Choosing Wisely Campaign - perhaps in the last paragraph - "do not prescribe lipid lowering medication for individuals with limited life expectancy" - https://www.choosingwisely.org/clinician-lists/
Author Response
That sounds like a great suggestion. I will update the manuscript with this.
Best,
Elizabeth Goodman
Reviewer 2 Report
introduction
The authors are truly clear to point out why they want to do this research.
- In line 32 there is an additional “)” in reference 2.
- The article should have been more addressed about the safety of high doses
Case report
The medical history of the case is clearly indicated, but it may be necessary to increase the period of the patient takes twice the dose of atorvastatin
Discussion
- There is an additional “.” in 104.
- Because of the dosage of drugs (the patient was noted to be taking atorvastatin 40 mg from two different bottles), there may be doubts in this conclusion.
- this is more likely to discuss the drug-related problem.
Author Response
Thank you for the correction for line 32 and line 104, they will be made!
We can definitely add a comment regarding the fact that higher doses/potencies are associated with increased risk.
Can you specify which aspect of the conclusion is in doubt so I can better address your concern?
It is unfortunately not known how long the patient had been taking double to prescribed dose of her statin.
Reviewer 3 Report
In this report, the authors have demonstrated that the need to conduct a risk benefit analysis prior to initiating new therapies in patients with limited life expectancy, including consideration of the potential for medication errors.
My comments and suggestions are shown as below:
- Baseline treatment status about hyperlipidemia and hypertension would be better to show a little more.
- The authors demonstrated the elevation of the data troponin-I, elevated AST and ALT, and a creatine kinase (CK). Then, it will be better if the authors demonstrated the change over time of those data (as shown in Table or Figure, if possible).
- The changes before and after treatment in sarcopenia or frail marker should be described a little more, such as change in Skeletal Muscle mass Index or hand grip measurement.
Author Response
I wanted to clarify what specifically you wanted to know about the baseline treatment of her HTN and HLD. Are you wanting to know the specific medication for HTN or her initial cholesterol levels perhaps? Let me know so I can best address your concern.
I will definitely try and add a table with the relevant lab data or give the specific values.
Unfortunately, neither the skeletal muscle mass nor the hand grip strength were available in this case.
Thank you.
Reviewer 4 Report
The case report has been written well.
Do you call the incident of rhabdomyolysis due to the medication as adverse drug reaction (ADR) or something else? That would be needed to connect your description and interpretation of this incident to the international knowledge of mediction management using appropriate scientific term. Please apply it throughout of your paper.
Also, your paper needs a conclusion under a separe subheading with the description of what should be done based on your observation in practice. Any implication of your finding would be appreciated.
Good luck.
Author Response
Thanks for your comments.
I think this is both an example of medication error (patient was taking double the prescribed dose) and an adverse drug reaction (while HMG-co-reductase myopathy can rarely be seen in the absence of a statin exposure, the vast majority of cases are associated with statin use. ) Is this what you would like me to clarify?
I think the conclusion is that our recommendation is that overall life expectancy and goals of care drive the decision making process with respect to the decision to initiate statin-so perhaps the final paragraph is what should go into a new section titled conclusions?
Thanks.
Round 2
Reviewer 2 Report
First, this entire manuscript is unfinished with fallowed statements.
- The table 1 missing the unit. it should not give as "(unit)'.
- The medication history is important in case report, it should include other medications used combined. Whether the patient is using other drugs at the same time? That will support your conclusion. And for how long the patient took twice dosage of atorvastatin, thought you have the answered that HMG-CoA and Necrotizing Myopathy is not dose dependent, but in this case, it will be an effective way to point out the safety of patient with Alzheimer’s dementia.
Author Response
Thank you for your review.
Table 1 now shows “units/liter” as opposed to “units” alone, this is the unit of measurement that is reported by the lab.
Unfortunately, we are unable to ascertain how long the medication dose was doubled, as that detail is not included in the clinical notes
Per my review the article already contains a linkage between the patient’s dementia and the increased risk of medication errors
This article was authored and reviewed by native English speakers, however I am open to any specific suggestions regarding English language changes required.
Thank you.
Reviewer 3 Report
Yes, if there were information about specific medication for HTN and/or her initial cholesterol levels. They might help the readers to distinguish whether overtreatment for elderly patient was performed or not.
I understood the other revisions.
Author Response
Thank you. I have added her full medication list to the manuscript. Her exact cholesterol levels are not available (as they were not integrated into our EMR at the time), however I have now included her estimate ASCVD risk based on her primary care physicians' note.
Reviewer 4 Report
nothing more.
Author Response
Thank you so much!